# Individualising Galvanic Vestibular Stimulation Further Improves Visuomotor Performance in Parkinson’s Disease

**DOI:** 10.3390/bioengineering12050523

**Published:** 2025-05-14

**Authors:** Anjali Menon, Madhini Vigneswaran, Tina Zhang, Varsha Sreenivasan, Christina Kim, Martin J. McKeown

**Affiliations:** 1Pacific Parkinson’s Research Centre, The University of British Columbia, Vancouver, BC V6T 2B5, Canada; amenon04@student.ubc.ca (A.M.); madhiniv@student.ubc.ca (M.V.); tina.zhang@ubc.ca (T.Z.); varsha.sreenivasan@ubc.ca (V.S.); christina.kim@ubc.ca (C.K.); 2MEDIC Foundation, Coquitlam, BC V3K 2Y9, Canada; 3Faculty of Medicine (Division of Neurology), The University of British Columbia, Vancouver, BC V6T 1Z3, Canada

**Keywords:** galvanic vestibular stimulation (GVS), individualised stimuli, Parkinson’s disease, two-pole/three-pole configuration

## Abstract

Impaired motor function is a defining characteristic of Parkinson’s disease (PD). Galvanic vestibular stimulation (GVS) has been proposed as a potential non-invasive intervention to enhance motor performance; however, its efficacy depends on both stimulation parameters and electrode configuration. In this study, we examined the effects of two-pole and three-pole GVS configurations, utilising different stimulation parameters, on motor performance in individuals with PD. Twelve participants with PD were administered eight distinct subthreshold amplitude-modulated GVS stimuli, along with sham stimulation, while performing a visuomotor target tracking task. Analysis of tracking error demonstrated substantial inter-individual variability in response to different stimuli and electrode configurations. While the three-pole configuration yielded superior motor performance in some cases, the two-pole configuration was more effective in others. The most effective overall stimulus across all subjects, characterised by an envelope frequency of 30 Hz and a carrier frequency of 110 Hz, improved motor performance by 25% relative to the sham stimulus. Moreover, tailoring the stimulation parameters to the individual further enhanced performance by an additional 24%. These findings suggest that GVS can yield significant motor improvements in individuals with PD. Furthermore, individualised optimisation of stimulation parameters, including the selection of the appropriate electrode configuration, may further enhance therapeutic efficacy.

## 1. Introduction

Parkinson’s disease (PD), a neurodegenerative disorder associated with a loss of dopaminergic neurons in the substantia nigra pars compacta, is characterised by impaired motor function, postural stability, and executive cognitive function [1,2]. It is becoming more prevalent, with the age-standardised incidence rate for PD in Canada of 53 per 100,000 for ages 40 and older in 2022 [3].

Dopaminergic medication (levodopa) remains the first-line treatment for Parkinson’s disease (PD) [4,5]. However, long-term use of levodopa is associated with the development of dyskinesias, medication “wearing-off” phenomena [6], and a potential decline in therapeutic efficacy over time [7]. Neurostimulation-based therapies serve as valuable adjuncts to pharmacological treatment [8], with deep brain stimulation (DBS) providing symptom relief in approximately 75% of patients [9]. However, DBS is inherently invasive and carries a risk of procedure-related complications [10]. Additionally, access to DBS is often constrained by long wait times and limited availability of specialised surgical centres [11,12]. Consequently, there is a need for minimally invasive, low-risk, and widely accessible therapeutic alternatives for PD management.

Some limitations associated with invasive brain stimulation may be addressed through non-invasive neuromodulation techniques, such as transcranial magnetic stimulation (TMS) and transcranial direct and alternating current stimulation (tDCS and tACS). These approaches have shown promising effects in improving motor symptoms in individuals with Parkinson’s disease (PD) [13,14,15,16]. However, their widespread clinical application is constrained by practical and regulatory challenges. TMS, for instance, requires large, specialised equipment and necessitates frequent in-clinic sessions, limiting accessibility. Additionally, while tDCS and tACS are more portable, they have yet to receive regulatory approval as established treatments for PD [17].

Galvanic vestibular stimulation (GVS) is an emerging non-invasive neuromodulation technique with minimal associated risk. By stimulating the vestibular system, GVS modulates basal ganglia activity through vestibular–striatal pathways, thereby influencing motor control [18,19]. Evidence suggests that GVS can improve various motor symptoms in Parkinson’s disease (PD) [20], including enhancements in fine motor performance [21] as well as stability and balance [22,23,24]. Notably, GVS can be delivered using a portable device, making it a potentially accessible and scalable therapeutic option.

GVS is typically characterised by the frequency and/or amplitude of the electrical current delivered, as well as the electrode configuration. Common stimulation modalities include direct current, sinusoidal current, and random noise stimulation [25]. Notably, the majority of previous GVS studies have utilised a two-pole electrode configuration [26,27,28], with limited investigation into alternative electrode arrangements, particularly in the context of Parkinson’s disease (PD). At higher current intensities, the three-pole configuration [27,28] has the potential to elicit anteroposterior head movements by engaging pathways associated with the anterior and posterior semicircular canals. In contrast, the conventional two-pole configuration can primarily induce lateral head movements by modulating neural pathways linked to the lateral semicircular canals [28]. Furthermore, most studies have employed a single GVS stimulus across all participants [29,30], without accounting for inter-individual variability in response. Consequently, the potential for personalised stimulus optimisation to maximise clinical benefits remains largely unexplored.

This study examined the effects of two-pole and three-pole electrode configurations, along with eight distinct amplitude-modulated (AM) stimuli, on visuomotor target tracking performance in individuals with Parkinson’s disease (PD). By analysing tracking error, we highlight the potential of personalised stimulation parameters to enhance motor performance in individuals with PD.

## 2. Methods

### 2.1. Study Participants

A total of *n* = 12 participants with PD participated in the study. The study protocol was approved by the Clinical Research Ethics Board at the University of British Columbia. All participants were patients at the Pacific Parkinson’s Research Centre and provided written informed consent prior to their participation in the study.

The inclusion criteria were as follows: (i) a diagnosis of idiopathic PD from a certified Movement Disorders specialist, (ii) age between 55 and 85 years, (iii) ability to walk independently, and (iv) sufficient fluency in English to complete questionnaires. Exclusion criteria included the following: (i) history of head surgery or neurosurgical procedures for PD such as DBS, (ii) history of any additional neurological disorders, (iii) dementia preventing informed consent, (iv) implantation of a cardiac pacemaker, wires or defibrillator, (v) frequent or severe headaches, and (vi) pregnancy.

Collected demographic data included age, sex, side most affected, disease duration, and medication dosage. Additionally, participants were evaluated with the Montreal Cognitive Assessment (MoCa) [31], the Movement Disorders Society Unified Parkinson’s Disease Rating Scale (MDS-UPDRS) Part III, and the Hoehn and Yahr scale [32].

### 2.2. Galvanic Vestibular Stimulation (GVS)

GVS was administered using a portable stimulator capable of delivering an arbitrary stimulus (MistyWest, Vancouver, BC, Canada). The two-pole configuration involved two patch electrodes placed on the mastoid processes, whereas the three-pole configuration involved two additional electrodes placed on the temples (Figure 1A, left). For each participant, the optimal current amplitude was determined using the staircase method to first determine the sensory threshold. Participants were stimulated starting with a base current of 0.1 mA, gradually increasing the current amplitude in steps of 0.1 mA, until the stimulation was perceived superficially (threshold) as local mild tingling of the skin under the electrodes. The optimal current amplitude was then calculated as 90% of this threshold value and was used in all subsequent stimulations. Optimal current amplitudes were determined independently for the two-pole and three-pole configurations.

Delivered stimuli were AM signals characterised by an envelope and a carrier frequency. Tested stimuli included the following 10 envelope-carrier frequency (in Hz) pairs: GVS1: [10, 30], GVS2: [10, 60], GVS3: [20, 60], GVS4: [20, 110], GVS5: [30, 75], GVS6: [50, 110] and GVS7: [70, 145], and GVS8: [30, 110]. These stimuli were chosen based on the results of a pilot study conducted previously, which demonstrated the best performance in the target tracking task (see below) [33]. Briefly, the carrier frequencies were chosen in the range of the typical resting firing rates observed for vestibular afferents [34]. The envelope frequencies were selected over a range comprising the alpha (8–12 Hz), beta (12–30 Hz), gamma (30–50 Hz), and high gamma (>50 Hz) bands of brain oscillations. The sampling rate of the stimuli was set to 60 Hz. Additionally, “sham” stimuli were included, where the electrodes were maintained in place, but participants received no stimulus (zero current).

### 2.3. Target Tracking Task

Participants were evaluated on a visuomotor target tracking task. This task was designed to assess fine motor skills through cursor movement and was programmed with MATLAB (R2022a) and Psychtoolbox-3. Participants were seated approximately 24 inches from a visual display screen (Asus Zenbook RGB, display size 15 inches, refresh rate 60 Hz). The task began with a 2 × 5 grid (10 squares) being displayed on the screen, where each square was associated with one of the ten stimuli described above (eight GVS and two sham stimuli). Participants then clicked on a square, which opened a second screen displaying a target ring and the cursor (a circle). Participants were instructed to track the target ring on the screen with the cursor using their dominant hand. The target ring moved along a Lissajous curve invisible to the participant (sampling rate 60 Hz; X frequency: 0.075 Hz, Y frequency: 0.1 Hz). Additionally, the cursor itself was perturbed by a smaller Lissajous curve (sampling rate 60 Hz; X frequency: 0.14 Hz, Y frequency: 0.21 Hz), adding noise. The goal of the participants was to maintain the cursor within the target ring as accurately as possible. The square that the participant chose determined the stimulus delivered. At the end of the trial, participants were directed back to the screen with the grid. They could now select a different square to begin another trial. Participants performed 10 trials of this task, moving through all 10 squares on the grid. For the first two trials, participants were directed to select squares corresponding to the sham stimulus. For the remaining trials, participants could select a square in any order. Therefore, the stimulus delivered was based on participants’ selection, and the order of stimulus delivery was different for each participant. Additionally, participants were blinded to the stimulus they received. Each trial lasted 45 s (i.e., stimulation lasted for 45 s) and was followed by a 15 s break where no stimulus was delivered. The target tracking task was performed once each for the two-pole and three-pole configurations.

### 2.4. Error Analyses

#### 2.4.1. Tracking Error

To compute tracking error, the Lissajous curves in the X and Y direction (and their Hilbert transforms) were first linearly regressed out from participants’ X and Y cursor positions, respectively. The residuals in the X ($Err_{X}$) and Y ($Err_{Y}$) directions were analysed as measures of tracking error. A time course of tracking error for each GVS was computed as(1)$$Err_{R}\left(i\right)\;=\;\sqrt{Err_{X}{\left(i\right)}^{2}+\;Err_{Y}{\left(i\right)}^{2}},$$
where *i* indicates a time point. The error time course was further segmented to obtain “sub-trials”. Specifically, the error time course was segmented into a length equal to one-quarter of the period corresponding to the target Lissajous Y frequency (0.1 Hz; sub-trial segment length: 2.5 s). This resulted in 18 sub-trials. The average tracking error for each sub-trial was computed as(2)$${Err}_{R}=\frac{1}{n}\sum_{i=1}^{n}{Err}_{R}(i),$$
where *n* is the number of time points. All error metrics were computed independently for each GVS for the two-pole and three-pole configurations.

The difference in tracking error between the two-pole and three-pole configurations was computed as(3)$$\Delta Err\;=\;\frac{1}{m}\sum_{i=1}^{m}\left({Err}_{R}^{two-pole}(i)\;-\;{Err}_{R}^{three-pole}(i\right)),$$
where *m* indicates the number of sub-trials. Negative values indicate better performance (lower error) with the two-pole configuration compared to the three-pole configuration.

The “individualised stimulus” was identified as the one that produced the least average tracking error (${Err}_{R}$), across all GVS and pole configurations for that individual. 

#### 2.4.2. Performance Improvement

The percentage performance improvement with each GVS over the sham stimulus was computed as(4)$${PI}^{GVS}=100\times \frac{{Err}_{R}^{SHAM}-{Err}_{R}^{GVS}}{{Err}_{R}^{GVS}},$$
where ${Err}_{R}^{SHAM}$ is the average tracking error with the sham stimulus and ${Err}_{R}^{GVS}$ is the average tracking error with any other GVS (GVS1, GVS, …, GVS8), averaged across the two-pole and three-pole conditions.

The percentage performance improvement of the individualised stimulus over every other GVS was computed as(5)$${PI}^{Ind}=100\times \frac{{Err}_{R}^{GVS}-{Err}_{R}^{Ind}}{{Err}_{R}^{Ind}},$$
where ${Err}_{R}^{Ind}$ is the average tracking error with the individualised GVS and ${Err}_{R}^{GVS}$ is the average tracking error with any other GVS (GVS1, GVS, …, GVS8), averaged across two-pole and three-pole conditions.

### 2.5. Statistical Analyses

The descriptive statistics used include mean and standard error of the mean. Since the sub-trials are not independent, a linear mixed effects model with sub-trials (Figure 1B and Figure 2A) and subjects (Figure 1B) included as random effects were used to test for significant differences between the two-pole configuration and the three-pole configuration. For each participant (Figure 2A), the mean ΔErr across sub-trials (and associated standard error) for each GVS is reported.

For all analyses, a significance level of 0.05 was used. False Discovery Rate (FDR) correction was conducted using the Benjamini–Hochberg method [35]. FDR-adjusted *p*-values [36] are reported.

### 2.6. Data Exclusion

Of the 12 participants recruited for the study, the target tracking task data from two participants (3P05 and 3P10) could not be recorded due to unanticipated technical issues. These subjects were excluded from all analyses. Therefore, analyses of the target tracking task included *n* = 10 participants.

## 3. Results

### 3.1. Demographics and Clinical Characteristics

The study enrolled 12 individuals with PD (4 females). The mean (standard deviation [SD]) age of the participants was 74.5 ± 6.6 years. Of the 12 participants, 10 exhibited greater disease impact on the left side. The mean (SD) UPDRS score was 46.6 ± 9.8 (Table 1). Cognitive assessments (MoCA) suggested mild cognitive impairment among the participants (Table 1).

### 3.2. Tracking Error as a Function of Stimulation Parameters

We tested participants on a visuomotor target tracking task. Briefly, participants tracked a target ring moving along a Lissajous curve (Figure 1A, right) with their cursor to maintain the position of the cursor within the target. Participants performed ten trials, eight trials with GVS (eight unique AM stimuli; Section 2.2) and two sham trials, each for the two-pole and three-pole configurations. Following completion of the task, the error between the target Lissajous curve and the path traced by participants was calculated (Section 2.4).

The mean tracking error ${Err}_{R}$ across all stimuli delivered was not significantly different between the two-pole and three-pole configurations (*p* > 0.05; Figure 1B, left). Investigating the tracking error for each stimulus independently, we observed no significant difference between two-pole and three-pole configurations (*p* > 0.05; Figure 1B, right) for any stimulus. Moreover, tracking error, while variable, was not markedly different across stimuli.

Further, we analysed the effect of each stimulus on tracking error, ${Err}_{R}$, independently for every participant. We observed that tracking error varied considerably not only across the different stimuli, but also based on the pole configuration (Figure 2A). Specifically, the difference in tracking error between the two-pole and three-pole configuration, ΔErr (Section 2.4), revealed that for most participants, some stimuli resulted in better performance with the two-pole configuration (negative ΔErr) whereas others resulted in better performance with the three-pole configuration (positive ΔErr; Figure 2A).

We next assessed potential performance improvements of all GVS over the sham stimulus. Overall, across all participants and pole configurations, GVS8 (envelope frequency 30 Hz, carrier frequency 110 Hz) produced the greatest percentage performance improvement over the sham stimulus (25%, Table 2, Section 2.4), and may be considered the best overall stimulus. However, with the stimulus that resulted in the least ${Err}_{R}$, the “individualised stimulus”, we observed that a further improvement of 24% over the best overall stimulus may be achieved (Table 2). Moreover, compared to the sham stimulus, the individualised stimulus reduced average tracking error by 51% across all participants (Figure 2B, *p* = 0.001, Wilcoxon signed-rank test). Additionally, 7/10 participants had individualised stimuli in the three-pole configuration.

## 4. Discussion

In this study, we investigated the effects of two-pole and three-pole GVS and different AM stimuli on performance on a visuomotor target tracking task in participants with PD. Across all participants tested, we observed no overall differences in tracking error in the visuomotor target tracking task between two-pole and three-pole configurations. However, participant-wise analysis of data revealed marked variability across responses to different stimuli, with the individualised stimuli demonstrating a potential of a 24% improvement in performance over the best overall stimulus (GVS8—envelope frequency 30 Hz; carrier frequency 110 Hz) and a 51% improvement over the sham stimulus. Our results suggest that stimulation parameters, when individualised, have the potential to significantly and positively affect motor performance in PD.

We explored several different AM stimuli as well as different electrode configurations. We chose AM stimuli with carrier frequencies in the range of the resting firing rates for vestibular afferents [34] and envelope frequencies in the alpha, beta, gamma, and high gamma bands of brain oscillations. While our findings are also aligned with studies investigating frequency-specific impacts on response time and response variability across stimuli [37], it is clear that the full range of potentially useful stimulus waveforms remains largely unexplored.

The neurophysiological mechanisms underlying the effects of GVS are still unclear. Possible mechanisms include influencing systems-level oscillatory rhythms, especially in basal ganglia–thalamocortical networks [38]. GVS may affect functional brain network connectivity [39,40], with potential significant effects on motor vigour in PD [41,42].

To the best of our knowledge, this is the first study to investigate the effects of three-pole GVS on motor performance in participants with PD. At the vestibular nuclei level, there is some indication of anatomical segregation of the different semicircular canals. The ascending branches of the superior and lateral canals terminate more rostrally in the superior vestibular nucleus, while the posterior canal afferents terminate more medially [43]. However, the downstream effects at the basal ganglia and cortical levels of such segregation are largely unknown. In our study, 7/10 subjects had individualised stimuli that included the three-pole configuration, suggesting that electrode configuration should also be considered in addition to waveform stimulation optimisation.

There are several limitations to our study. All conclusions are based on a small sample of 10 participants. It is possible that the best overall stimulus (GVS8—envelope frequency 30 Hz; carrier frequency 110 Hz) and the associated performance improvements identified with the current sample size may not be robust and could vary. Nevertheless, we believe our central thesis that substantial improvement in the efficacy of GVS can be realised with individualised stimuli and electrode configurations will be robust to sample size. Additionally, we limited our study to a fixed set of AM stimuli; however, to accurately identify optimal and individualised stimuli, a more comprehensive repertoire of other classes of stimuli, such as sinusoidal, multisine, or noisy GVS, may be necessary.

Our study demonstrates that GVS can result in significant motor improvements, especially with individualised stimuli and electrode arrangements. GVS is a safe and non-invasive neuromodulation technique that may be easily and quickly implemented in clinical settings. GVS equipment is compact and easy-to-operate, consisting of a stimulator, electrodes and conductive gel. Moreover, portable stimulators such as the one used in this study provide added benefits of evaluating the effects of GVS in a wide range of mobility assessments, such as gait and balance tasks that typically require patient movement. GVS can therefore be a scalable and effective intervention for improving motor symptoms associated with PD.

## Figures and Tables

**Figure 1 bioengineering-12-00523-f001:**
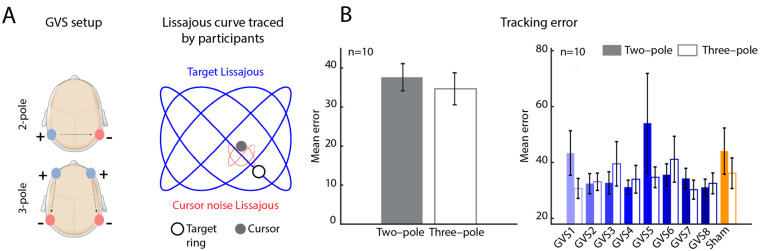
Study setup and tracking error comparing two-pole and three-pole configurations. (**A**). (Left) Schematic representation of GVS setup showing the placement of electrodes for the two-pole (top) and three-pole (bottom) configurations. (Right) Lissajous curve traced by participants in the visuomotor target tracking task (blue). Cursor noise Lissajous is indicated in red along with the target ring (black circle) and cursor (grey circle). (**B**). (Left) Median tracking error pooled across all participants and stimuli for the two-pole (black) and three-pole (grey) configurations. (Right) Median tracking error for each stimulus, pooled across all participants for the two-pole (filled bars) and three-pole (open bars) configurations. Error bars indicate the standard error of the median. GVS, galvanic vestibular stimulation; n, number of participants.

**Figure 2 bioengineering-12-00523-f002:**
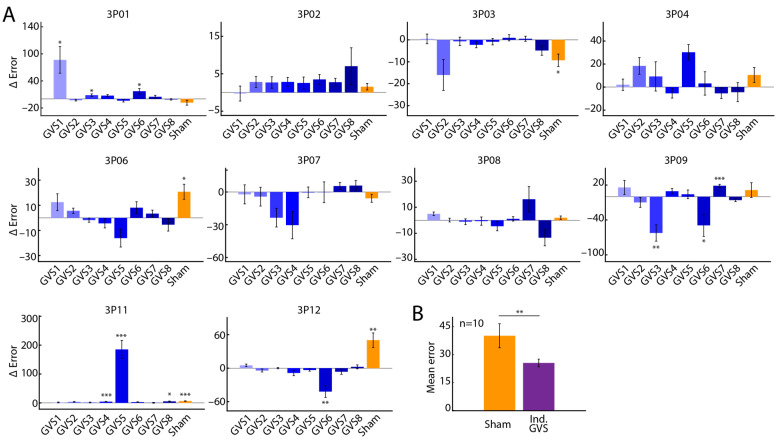
Participant data showing tracking error difference between the two-pole and three-pole configurations. (**A**). Mean difference in tracking error between the two-pole and three-pole configurations for each stimulus (blue filled bars), including sham (yellow filled bar). (**B**). Mean tracking error for the sham stimulus (yellow filled bar) and the individualised stimulus (purple filled bar). Error bars indicate the standard error of the mean. * *p* < 0.05; ** 0.01 < *p* < 0.05; *** *p* < 0.001. GVS, galvanic vestibular stimulation; n, number of participants.

**Table 1 bioengineering-12-00523-t001:** Demographics and clinical characteristics.

Characteristic	
**Age, years**	
Mean (SD)	74.5 (6.6)
**Gender, n**	
Female	4
Male	8
**Side affected, n**	
Left	10
Right	2
**MoCA score**	
Mean (SD)	26.4 (4.2)
**UPDRS III score (total)**	
Mean (SD)	46.6 (9.8)

MoCA, Montreal Cognitive Assessment; n, number of participants; SD, standard deviation; UPDRS, Unified Parkinson’s Disease Rating Scale.

**Table 2 bioengineering-12-00523-t002:** Improvements of individualised GVS over the overall best GVS.

Stimulus	Improvement with Individualised GVS (%)	Improvement with GVS over Sham (%)
GVS1	47	12
GVS2	28	19
GVS3	37	13
GVS4	27	23
GVS5	87	7
GVS6	45	4
GVS7	25	21
GVS8	24	25
SHAM	51	0

GVS, galvanic vestibular stimulation.

## Data Availability

Data supporting the findings of this study are available at https://doi.org/10.5683/SP3/3WM1SP (accessed on 26 March 2025).

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
