# Peer review of "Individualising Galvanic Vestibular Stimulation Further Improves Visuomotor Performance in Parkinson’s Disease"

_bioengineering, 2025, doi:10.3390/bioengineering12050523_

Round 1

Reviewer 1 Report

Comments and Suggestions for Authors

This paper can be accepted.

Reviewer 2 Report

Comments and Suggestions for Authors

The authors explore the possibilities of using galvanic vestibular stimulation for improving visuomotor performance in Parkinson's disease.

  1. The sample size (n=10) is rather small for the scope of this study. Consider acknowledging this more explicitly in the limitations section and discussing how it might affect the generalisability of your findings.
  2. Include details about whether the order of stimulus delivery was randomised.
  3. Table 1 is incorrectly labelled as Table 12.
  4. The study lacks a control group; please acknowledge this in the limitation section.
  5. Also, consider addressing practical aspects of implementing GVS in clinical settings.

Reviewer 3 Report

Comments and Suggestions for Authors

The presented work focuses on the study of what effect individual galvanic vestibular stimulation has on motor function in Parkinson's patients. The paper presents the result of a comparative analysis of two-pole and three-pole electrode configurations. The parameters of amplitude-modulated stimulation are also investigated in the paper. Visuomotor tracking experiments were performed with 12 participants. The result was very noteworthy: with a carrier frequency of 110 Hz the accuracy of movements was improved by 25%, while the envelope frequency was equal to 30 Hz. If we take into account the individual characteristics of patients, it is possible to achieve an increase in efficiency by 24%.  

Due to the relevance of the development of methods of correction of motor disorders due to Parkinson's disease. It should be noted that in the work special attention is paid to the consideration of personalized approach with consideration of different electrode configurations with demonstration of response to stimulation.

The method of statistical analysis was used to conduct the study, and the order of the experiment was arranged in such a way as to avoid side factors in the course of multiple comparisons. The structure of the article is organized competently and corresponds to the type of academic scientific articles. Thanks to the presented graphs and tables, the perception of data is simplified.

In my opinion, the paper reaches a very important conclusion of confirming the therapeutic effect on the patient by means of galvanic vestibular stimulation. The advantage of three-pole configuration for a part of patients is undoubted, while the authors focus on individual selection of stimulation parameters to achieve the best effect. I think that the work will be useful for neurologists and rehabilitologists.

I consider the article submitted for review to be of high quality, possessing original research and recommendations for the treatment of patients suffering from Parkinson's disease. I believe that the work deserves to be published in the scientific journal.
